



# Daily soil temperature modeling improved by integrating observed snow cover and estimated soil moisture in the U.S. Great Plains

Haidong Zhao[1], Gretchen F. Sassenrath[2,3], Mary Beth Kirkham[3], Nenghan Wan[1], Xiaomao Lin[1*]

[1]Department of Agronomy, Kansas Climate Center, Kansas State University, Manhattan, Kansas, USA

[2]Southeast Research and Extension Center, Parsons, Kansas, USA

[3]Department of Agronomy, Kansas State University, Manhattan, Kansas, USA

10    *Correspondence to*: X. Lin (xlin@ksu.edu)





**Abstract**

15 Soil temperature ($T_s$) plays a critical role in land-surface hydrological processes and agricultural

ecosystems. However, soil temperature data are limited in both temporal and spatial scales due to the

configuration of early weather station networks in the U.S. Great Plains. Here, we examined an

empirical model (EM02) for predicting daily soil temperature ($T_s$) at the 10 cm depth across Nebraska,

Kansas, Oklahoma, and parts of Texas that comprise the U.S. winter wheat belt. An improved empirical

20 model (iEM02) was developed and calibrated using available historical climate data prior to 2015 from

87 weather stations. The calibrated models were then evaluated independently using the latest 5-year

observations from 2015 to 2019. Our results suggested that the iEM02 had, on average, an improved

root mean square error (RMSE) of 0.6ºC for 87 stations when compared to the original EM02 model.

Specifically, after incorporating changes in soil moisture and daily snow depth, the improved model

was 50% more accurate as demonstrated by the decrease in RMSE. We conclude that in the U.S. Great

Plains the iEM02 model can better estimate soil temperature at the surface soil layer where most

hydrological and biological processes occur. Both seasonal and spatial improvements made in the

improved model suggest that it can provide a daily soil temperature modeling tool that overcomes the

deficiencies of soil temperature data used in assessments of climatic changes, hydrological modeling,

and winter wheat production in the U.S. Great Plains.



## 1 Introduction

A reliable estimate of soil temperature ($T_s$) is useful to understand agricultural ecological systems, hydrological processes, and land-atmosphere interactions (Lembrechts et al., 2020; Qi et al., 2016; Zhang et al., 2018) due to the fact that $T_s$ governs physical, chemical, and biological processes of the

soil and interactions between the atmosphere and land-surface  (Smith, 2000; Soong et al., 2020). In particular, $T_s$ has been widely used for a better understanding of changes in soil moisture (Lakshmi et al., 2003), the ecosystem carbon balance (Goulden et al., 1998), and the nitrogen mineralization process (Persson and Wirén, 1995) although a larger prevalence of air temperature observations are available as a soil temperature proxy. From a practical perspective, $T_s$ is critical for agricultural system models such

as the crop environmental resource synthesis (CERES) models to assess the impacts of extreme climate on crop production and stress tolerance, thereby allowing producers to better prepare for proactive and reactive field management (Bergjord et al., 2008; Persson et al., 2017; Williams et al., 1989). Frequent extreme climate events such as spring freezes and summer heat stress can impact winter wheat [*Triticum aestivum L.*] growth and development, reducing grain yields by more than 7% in the U.S.

winter wheat belt (Tack et al., 2015; Paulsen and Heyne, 1983). These effects are also modulated through land-surface interaction processes (Hillel, 1998; Araghi et al., 2017).

To improve the accuracy of crop management modeling, a bare soil temperature ($T_s$) at the 10 cm depth, a standard soil temperature variable, has commonly been considered as a more direct and useful

variable than air temperature ($T_a$) measured at 1.5 or 2 m height in crop phenology (Onwuka and Mang, 2018), plant photosynthesis and soil respiration (Meyer et al., 2018; Wu and Jansson, 2013), plant



nutrient uptake (Yan et al., 2012), and estimate of crop production (Araghi et al., 2017; Hillel, 1998).

There are many $T_s$ modeling techniques mostly based on the land-surface interaction process (Qi et al.,

2019; Yener et al., 2017). Most $T_s$ models are rooted in theories of soil heat exchange and surface

energy balance (Rankinen et al., 2004; Nobel and Geller, 1987; Chalhoub et al., 2017). The theory-

based simulation for surface energy balance usually includes solar radiation (incoming and outgoing),

infrared radiation (absorbed and reflected), turbulent flux energy (latent heat and sensible heat), and net

ground heat flux through the ground surface into soil layers thermodynamically (Mihalakakou et al.,

1997; Chalhoub et al., 2017). Obviously, the energy-balance based model usually requires more detailed

near-surface and soil variables such as turbulent flux quantities (sensible heat flux and latent heat flux)

to make the model reliable and accurate; however, determining quality turbulent flux quantities is not a

trivial task (Kutikoff et al., 2021; Dhungel et al., 2021). In addition, seasonal variations of soil thermal

conductivity and underestimates of actual evapotranspiration usually lead to overestimated surface soil

temperatures (Bittelli et al., 2008). Therefore, simpler empirical models with fewer dynamic processes

needed to predict $T_s$ have been explored (Zheng et al., 1993; Plauborg, 2002; Liang et al., 2014;

Badache et al., 2016; Kang et al., 2000). However, these empirical models might result in relatively

large estimated errors of over 2°C due to the lack of details about physical process such as uncertainties

of the soil volumetric heat capacity and thermal conductivity (Badía et al., 2017). For example, the

volumetric heat capacity was higher for a clay soil, which ranged between 1.48 and 3.54 MJ m$^{-3}$ $^o$C$^{-1}$,

than for a sand soil, which ranged between 1.09 and 3.04 MJ m$^{-3}$ $^o$C$^{-1}$ when the soil moisture content

was between 0 to 0.25 kg kg$^{-1}$ (Abu-Hamdeh, 2003).

Currently, the U.S. Department of Agriculture (USDA) provides a high-resolution Gridded Soil Survey

Geographic (gSSURGO) Database product (https://gdg.sc.egov.usda.gov/) that includes static soil

physical property data at 10 km resolution (USDA NRCS, 2013). The gSSURGO data facilitate $T_s$

modeling, especially for better performance in large-scale $T_s$ modeling due to its spatial variations in

soil properties and soil moisture (USDA NRCS, 2013). These datasets have been widely used in the

estimation of root-zone soil water content (Miller et al., 2018) and sub-surface hydrologic properties

(Dirmeyer and Norton, 2018). The empirical model proposed by Plauborg (2002) performed better than

energy-balance based models when applied in the U.S. Great Plains for the last five years. Due to the

lack of information about static soil properties on a large scale one or two decades ago, either over- or

underestimates of $T_s$ occurred, which gave a large deviations in the assessment of crop stress and crop

production (Gupta et al., 1990; Stone et al., 1999).

Recent studies have shown that estimated soil temperature usually deviates from observed soil

temperature in the winter due to snow cover, frozen soil, and wide spatial and temporal heterogeneity in

frozen soil properties (Nagare et al., 2012; Zhang et al., 2008; Rankinen et al., 2004). The impact of

snow cover on soil temperature has been investigated (Rankinen et al., 2004) and is partially accounted

for by incorporating correcting factors in land-surface modeling as well as ecosystem models (Zhang et

al., 2008) and soil and water assessment tools (SWAT) (Qi et al.2019). For both empirically and

physically-based soil temperature modules embedded in SWAT, the predictions of soil temperature in

regions with thick snow cover seldom agree with field measurements in winter (Qi et al. 2019).

In the U.S. Great Plains, there has been increasing interest in improving hydrological process modeling

of surface water and groundwater due to the Ogallala aquifer's depletion in recent decades (Haacker et

al., 2019). However, observed soil temperature information has been provided by the automated

weather station networks in this region that was commissioned in the late 1980s and early 1990s (Brock

and Crawford, 1995). Not only were there few continuous observations for $T_s$ earlier than the 1990s,

these automated weather station networks also had limited stations in each state of the U.S. Great

Plains. Such a lack of reliable soil temperature data both spatially and temporally makes the long-term

assessment of water resources, crop phenology, and crop production modeling difficult.

The objectives of this study include: (1) develop a robust $T_s$ model using limited surface climate

variables by integrating soil moisture estimates dynamically as well as snow depth observations; (2)

demonstrate the error to contributions in soil temperature modeling; and (3) evaluate the performance of

an improved model to predict $T_s$ compared to current models. The datasets and methods used are

described in section 2. Section 3 provides modeling results and conclusions are presented in section 4.

## 2 Datasets and Methods

### 2.1 Weather stations and datasets

The spatial domain of this study covers the winter wheat belt in the U.S. Great Plains, comprising the

states of Nebraska (NE), Kansas (KS), Oklahoma (OK), and part of Texas (TX) where soil texture and

bulk density vary (Fig. 1). In this study, three surface climate datasets were obtained from the

Automated Weather Data Networks (AWDN) (https://hprcc.unl.edu/awdn/), commissioned in the 1980s



for Nebraska and Kansas. The Oklahoma Mesonet is a daily climate data source for Oklahoma, which

started in the 1990s (http://www.mesonet.org/). For Texas, we selected the Soil and Climate Analysis

Network for its daily climate observations (https://www.wcc.nrcs.usda.gov/scan/) due to limited quality

data available in its automated weather station network. The selected stations included 26 in NE, 8 in

KS, 44 in OK, and 9 in TX. The selection of these 87 stations was based on the completeness of climate

data and data length (at least longer than a continuous 15-year periods). In addition to the weather

station datasets, soil datasets providing soil attributes and characteristics were obtained from the

standard USDA-NRCS Soil Survey Geographic (gSSURGO) Database product

(https://gdg.sc.egov.usda.gov/), in which soil bulk density ($\rho_b$, g cm$^{-3}$), soil organic matter ($f_{OM}$, %),

sand ($f_{sa}$, %), clay ($f_{cl}$, %), silt ($f_{sl}$, %) contents, soil porosity (Ø, %), and soil surface albedo ($\alpha$, -) were

used for all weather stations. Note that all symbols and corresponding descriptions for variables used in

this study are listed in the Table A1 (see the Appendix). The snow depth data were taken from the daily

Global Historical Climatology Network (GHCN) (Menne et al., 2009; Lin et al., 2017). Detailed dataset

sources and data variables used in each dataset are shown in Table A2 (see the Appendix).

## 2.2 Soil temperature models

**2.2.1 Empirical model**

There are two common soil temperature models: empirical and process-based. After examining both

types of models for our study region, the current empirical model was selected because it was more

accurate than the process-based model in this area. Plauborg (2002) developed a statistical soil

temperature ($T_s$, ºC) model based on the current and previous two-day air temperatures ($T_a$, ºC), annual





and semi-annual cycles in the soil temperature fluctuations, and a daily soil temperature offset at a

specific site, as shown in Eq. (1) (called EM02, thereafter):

$$T_{s,j} = \gamma + \alpha_0 T_{a,j} + \alpha_1 T_{a,j-1} + \alpha_2 T_{a,j-2} + \beta_1 sin(\omega j) + \delta_1 cos(\omega j) + \beta_2 sin(2\omega j) + \delta_2 cos(2\omega j)$$

(1)

where $\gamma$ is an offset constant (°C) and coefficients $\alpha_0$, $\alpha_1$, and $\alpha_2$ are dimensionless. The units of the

coefficients $\beta_1$, $\beta_2$, $\delta_1$, and $\delta_2$ are Celsius (°C). The $j$ and $\omega$ denote day of the year and annual frequency

($2\pi/365$ or $2\pi/366$ in leap years) in an annual soil temperature signal.

### 2.2.2 Improved empirical model

The improved model, based on the EM02, was developed through the following three steps: (1)

prolonging the time window of $T_a$ to include one extra prior day $T_a$; (2) constructing a new fictive

environmental temperature ($T_{env}$, °C) defined as a function of air temperature and surface skin

temperature ($T_{sfc}$, °C) (Williams et al., 1984) utilizing $T_{env}$ to replace the original $T_a$; and most

importantly (3), incorporating site-specific daily soil thermal diffusivity and snow depth (Fig. 2). This

improved empirical model (iEM02) can be described by Eqs. (2-6):

$T_{s,j} = \big(\gamma + \alpha_0 T_{env,j} + \alpha_1 T_{env,j-1} + \alpha_2 T_{env,j-2} + \alpha_3 T_{env,j-3} + \beta_1 sin(\omega j) + \delta_1 cos(\omega j) +$

$\beta_2 sin(2\omega j) + \delta_2 cos(2\omega j)\big) \times f\big(D_{S,j}\big) \times DR_{eff,j}$        (2)

$T_{env,j} = \beta T_{a,j} + (1-\beta)T_{sfc,j}$        (3)

$T_{sfc,j} = (1-\alpha)\left(\bar{T}_{a,j} + \big(\bar{T}_{max,j} - \bar{T}_{a,j}\big)\sqrt{\frac{R_{s,j}}{33.5}}\right) + \alpha T_{sfc,j-1}$        (4)

$f\big(D_{S,j}\big) = exp\,(-f_S D_{S,j})$        (5)





$$DR_{eff,j} = exp\left(k_0\sqrt{-h\frac{\pi}{k_{s,j}\,p}}\right) \qquad (6)$$

where $\beta$ refers to the weighting coefficient for air temperature $T_a$ (-). The $T_{sfc}$ in Eq. (3) was estimated

iteratively from the three-day running average of daily air temperature ($\bar{T}_a$), daily maximum

temperature ($\bar{T}_{max}$, ºC), and daily solar radiation ($R_s$, MJ m⁻² d⁻¹). The $\alpha$ denotes soil surface albedo (-)

and initial $T_{sfc,j-1}$ was set as annual mean $T_a$ in Eq. (4). The constant 33.5 is an empirical constant (MJ m⁻

² d⁻¹) (Williams et al., 1984). The function of snow cover on the jᵗʰ day is given as $f(D_{S,j})$ and was

introduced based on the work of Rankinen et al., (2004). The $f_S$ and $D_S$ are empirical soil heat damping

parameters (m⁻¹) and snow depth (m). The damping ratio of soil at the soil depth of $h$ ($h = 0.1$ m in this

study) is $DR_{eff,j}$ (Rosenberg et al., 1983). The weighting coefficient for the damping ratio (-) is $k_0$. The $p$

represents the period (365 days or 366 days in a leap year) in an annual cycle. The thermal diffusivity $k_{s,}$

$_j$ (m² s⁻¹) is equivalent to thermal conductivity ($\lambda$, W m⁻¹ K⁻¹) divided by volumetric heat capacity ($C$, J

m⁻³ K⁻¹) and reflects both the ability of soil to transfer heat and to change temperature when the heat is

supplied or dissipated. The estimate of thermal conductivity ($\lambda$) and volumetric heat capacity ($C$) can be

described by Eqs. (7-11) (Lu et al., 2014):

$$\lambda_j = \lambda_{dry} + exp\left(b_1 - \theta_j^{-b_2}\right) \qquad (7)$$

$$\lambda_{dry} = -0.56\emptyset + 0.51 \qquad (8)$$

$$b_1 = 1.97 f_{sa} + 1.87\rho_b - 1.36 f_{sa}\rho_b - 0.95 \qquad (9)$$

$$b_2 = 0.67 f_{cl} + 0.24 \qquad (10)$$

$$C_j = 1.92 \times 10^6 f_m + 2.51 \times 10^6 f_{OM} + 4.18 \times 10^6 \theta_j \qquad (11)$$





where $\lambda_{dry}$ is oven-dried soil thermal conductivity derived from a linear function of soil porosity (Ø, %).

Both $b_1$ and $b_2$ are the shape factors of the $\lambda$ curve that are estimated by soil texture components. Soil

water content is defined as $\theta_j$ on the $j^{th}$ day (cm$^3$ cm$^{-3}$) and was calculated by the soil water balance

model (Chalhoub et al., 2017). Briefly, the iEM02 operates on a daily time step as daily soil moisture is

a function of soil moisture storage capacity ($\theta^*$, mm), 24-hour precipitation ($P$, mm), and Penman-

Monteith reference evapotranspiration ($ET_0$, mm) and are estimated by Eqs. (12-15):

$$\theta_r = 0.026 + 0.005f_{cl} + 0.0158f_{OM} \tag{12}$$

$$\beta_{d,j} = 1 - \exp\left(-\frac{6.68\theta_j h}{(\theta_s - \theta_r)h}\right) \tag{13}$$

$$E_j = \begin{cases} P_j + \beta_{d,j}(ET_{0,j} - P_j) & P_j < ET_{0,j} \\ ET_{0,j} & P_j \geq ET_{0,j} \end{cases} \tag{14}$$

$$\theta_j h = \begin{cases} \theta_r h & \theta_j h \leq \theta_r h \\ \theta_{j-1}h + (P_{j-1} - E_{j-1}) & \theta_r h < \theta_j h < \theta^* h \\ \theta_s h & \theta_j h \geq \theta^* h \end{cases} \tag{15}$$

where $\theta_r$ and $\theta_s$ define residual and saturated volumetric soil water contents (cm$^3$ cm$^{-3}$). $\theta_s$ is assumed to

be equal to soil porosity while $\beta_{d,j}$ is a weighting coefficient for the difference between $ET_0$ (Allen et al.,

1998) and $P$ on the $j^{th}$ day (-). The initial soil water content ($\theta_{j-1}$) is assumed to be equal half of soil

porosity.

Climate observation data prior to the year 2015 were selected to calibrate the iEM02 for each station.

For NE, KS, and OK, daily soil temperature observations at each station had at least 10 years in daily

time series for calibrations. Datasets from TX had at least 4 years available for calibrations. Climate

variables used for calibration included air temperature, precipitation, snow depth, and solar radiation



daily observations and the site's static soil property. The optimal parameter values for each weather

station were estimated when a minimum root mean square (RMSE) between estimated and observed

soil temperature was achieved. These parameters for all 87 stations are listed in Table A3 (see the

Appendix).

### 2.3 iEM02 evaluation

In the datasets selected, all 87 station observations were longer than 15 years except for stations located

in Texas. The last five-year observations (2015 to 2019) were used to independently conduct model

validation for all 87 stations. The metrics used to evaluate model performance were root mean square

error (RMSE) and mean absolute error (MAE). Soil temperature modeling improvement was evaluated

by relative RMSE changes $[-\frac{100\left(RMSE_{improved}-RMSE_{original}\right)}{RMSE_{original}}]$ and by intercomparison between the fully

complete model and the reduced model.


## 3 Results and discussion

### 3.1 Improved empirical model (iEM02)

The iEM02 was evaluated from 2015 to 2019 for 87 weather stations. Soil temperature modeling using

different soil textures was improved in different ways in the iEM02 model (Fig. 3). The improvement of

soil temperature modeling improvement by relative RMSE changes was different for different sites. The

weather stations located in NE and KS as well as TX showed less improvement by introducing the air

temperature of $T_{a,j\text{-}3}$ compared to OK (Fig. 3a). The soil types in OK are more clay and silt compared to

NE and KS (Fig. 1). However, the improvement by using the fictive environmental temperature was





significant in northern areas of NE and KS but not in the southern area of OK and part of TX (Fig. 3b).

Overall, latitude-dominated air temperature should play a role in improving estimated soil temperature.

Most of the 87 stations achieved a 15% to 40% improvement in simulated soil temperature by

introducing air temperature $T_{a,\,j\text{-}3}$ and replacing $T_a$ with $T_{env}$. This improvement was in agreement with a

previous study (Dolschak et al., 2015). By incorporating changes in soil moisture and daily snow depth,

additional improvements in soil temperature simulation of up to 50% could be achieved (Fig. 3c)

compared to the original model EM02. It should be noted that there were fewer stations available in KS

and TX compared to NE and OK. Overall, integrating snow cover and soil moisture data in iEM02

improved the simulated soil temperature (Fig. 3).

**3.2 iEM02's parameters**

The parameters described in iEM02 for each weather station are indicative of soil temperature

sensitivities for each independent variable in Eq. (2) although strictly speaking, they are not

mathematical sensitivities (Fig. 4 & Table A2). For $T_{env}$, the current day $T_{env}$ was the most weighted as

expected (Fig. 4a). The parameters of $T_{env}$ for the prior day 1 to day 3 were relatively weak in terms of

absolute magnitudes due to autoregression properties in the soil temperature (Figs. 4b-d). Interestingly,

in the iEM02 model, the prior day 2 was negatively associated with soil temperature (Fig. 4c) which

cannot be interpreted by soil physical processes but in a more autoregressive sense in which the soil

temperature signals are superimposed. The periodic property embedded in iEM02 was two low-

frequency components (semi-annual and annual signals). Obviously, the annual signal strength

indicated by $\beta_1$ and $\delta_1$ was one-order stronger than the semi-annual signal strengths in soil temperature

(Fig. 4e-h). The result also suggested the strong $\beta_1$ and $\delta_1$ spatial contexts of the northern region (e.g., in

Nebraska and Kansas) were differently weighted than those from the southern region (e.g., in Oklahoma

and Texas). For the snow damping factor, the snow cover had a larger impact on soil temperature in the

northern region when compared to the southern region (Fig. 4i). However, the soil damping ratio factor

was relatively evenly distributed (Fig. 4j).

RMSE performance is shown in Figure 5 when the iEM02 was a complete model vs. the reduced model

iEM02 where one independent variable term was removed. When removing any one independent

variable, the modeled soil temperature RMSE increased from 110% to 130% (Fig. 5), indicating a 20%

drop in RMSE if one independent variable was removed in the iEM02 model. Specifically, the iEM02

model performance decreased (i.e., RMSE increased from 0.1 to 0.4°C) when the $\alpha_0$ term was removed

(Fig. 5, a-d). Unlike $\alpha_0$, removing the $\beta_1$ term was not as sensitive and gave an increase of 0.1-0.2°C

RMSE on average for all states in the region (Fig. 5, e-h). However, it is clear that the iEM02 model

was most sensitive to $\delta_1$. With the removal of $\delta_1$ from the complete iEM02 model, the RMSE increased

0.3-0.4°C for all four states (Fig. 5, i-l). Due to the location-dependency of the above coefficients,

further spatial interpolation of the iEM02 model would be beneficial to predict soil temperature for

irrigated agricultural areas without weather stations in the U.S. Great Plains and to improve water and

crop management modeling.

### 3.3 Spatial and temporal modeling performance

A graphical summary of how closely the modeled soil temperature agreed with the observed soil

temperature for each weather station is shown in Figure 6. Daily $T_s$ estimated in the iEM02 model

outperformed that in the original EM02 model for all 87 weather stations. For example, both mean

absolute error (MAE) and root mean square error (RMSE) were decreased on average by 0.6ºC when

the iEM02 model was used to estimate $T_s$. Individually, the improved model showed a less than 1.6ºC

RMSE for any individual station but 16% of the stations had larger than 2ºC RMSE in the original

EM02. In addition, we compared the performance of iEM02 against a recent energy-balance model

(Chalhoub et al., 2017). Our prediction of $T_s$ was improved by 1.2ºC RMSE compared to the energy-

balance model (not shown).

Spatial distributions of RMSE showed that the majority of weather stations had better performance in

Oklahoma with a mean RMSE of 1.9 and 1.1ºC for EM02 and iEM02, respectively, whereas Nebraska

had a RMSE of 2.1 and 1.3ºC for EM02 and iEM02, respectively. The different modeling performance

was associated with the soil heat transport process and how frequent snowfall could be observed in

Nebraska and Oklahoma Similar results were presented in a recent study by (Huang et al., 2017). On the

other hand, the high quality of weather data from the Oklahoma Mesonet considered to be the "gold

standard" for the statewide weather network (Lin et al., 2016) thus ensured quality of both model

calibrations and observed soil temperature in Oklahoma.

Seasonal $T_s$ indicated that iEM02 modeling was mostly improved in the spring season from 2ºC to

1.3ºC RMSE (Fig. 7a) but the original model EM02 showed the uncertainty was in good agreement

with the performance achieved in Plauborg (2002). All other seasons were improved in similar ways

from 1.8ºC to 1.2 or 1.3ºC RMSE. The improvement for all seasons could be attributed to introducing

soil diffusivity, which changed with daily soil moisture and snow cover, and this affected the soil

thermal conductivity (Rankinen et al., 2004; Zhang, 2005). Moreover, although modeling wintertime

soil temperature improved from 1.8ºC to 1.3ºC RMSE, which was the same as in the summer (Fig. 7),

the soil temperature located in more frequent snow-covered states, (e.g., Nebraska and Kansas), was

better improved when $T_{env}$ and snow depth were introduced (Rankinen et al., 2004; Dutta et al., 2018).

Since precipitation gradients exist in the U.S. Great Plains from western to eastern regions (Evett et al.,

2020), three subregions were classified for each state as western (100ºW towards west), central

(between 97º and 100ºW), and eastern ( 97ºW towards east). Figure 8 displays the time series of EM02

modeled, iEM02 modeled, and observed soil temperatures only covering winter wheat growing seasons

(October 1 to June 30) for four growing seasons from 2015 to 2019 (validation periods) in Nebraska and

Kansas. All subregions in Nebraska and Kansas showed improvement when using the iEM02 model

(Fig. 8). Similarly, the iEM02 improved the RMSE during four growing seasons in Oklahoma and

Texas (Fig. 9). The EM02 model had the best performance in Oklahoma with a mean RMSE of 1.0ºC,

while the mean RMSE in Kansas was 1.4ºC in EM02. Soil temperatures estimated by iEM02 had

approximately a 0.3 to 0.4ºC RMSE (Figs. 8 and 9). In addition, larger improvements by iEM02 were

observed in most subregions during wintertime, which would be beneficial for modeling accurately

winter wheat yields and potential yields (Persson et al., 2017).


## 4. Conclusion





The primary intention of this work was to develop an improved soil temperature model for the U.S. Great Plains that can predict soil temperature by using common weather station variables as inputs. The improved empirical model (iEM02) integrated soil thermal diffusivity and snow cover factors, and they significantly improved the estimate of soil temperature for 87 weather stations in the U.S. Great Plains that were studied. Specifically, after incorporating changes in soil moisture and daily snow depth, the improved model showed a near 50% gain in performance in terms of RMSE decrease in the improved model compared to the original model. The value of RMSE across 87 stations was 0.6°C lower on average than the original model from 2015 to 2019. We concluded that the iEM02 model can estimate better soil temperature at the surface soil layer where most hydrological and biological processes occur. Both seasonal and spatial improvements made in the improved model demonstrated the robustness of the iEM02 model, suggesting this improved model can provide a reliable simulation of soil temperature to use in modeling hydrological process and crop production in the U.S. Great Plains.



## Appendix

**Table1 A1.** Table of symbols and corresponding descriptions used in this paper.

| Symbols | Descriptions | Units |
|---|---|---|
| $\alpha$ | soil surface albedo | (-) |
| $\alpha_0, \alpha_1, \alpha_2, \alpha_3$ | empirical parameters of air temperature to estimate soil temperature | (-) |
| $\beta$ | empirical parameter of air temperature to calculate environmental temperature | (-) |
| $\beta_1, \beta_2$ | empirical parameters of sine wave to estimate soil temperature | (ºC) |
| $\beta_d$ | empirical parameter of evapotranspiration for actual evapotranspiration | (-) |
| $\delta_1, \delta_2$ | empirical parameters of cosine wave to estimate soil temperature | (ºC) |
| $\gamma$ | offset constant | (ºC) |
| $\lambda$ | soil thermal conductivity | (W m$^{-1}$ K$^{-1}$) |
| $\lambda_{dry}$ | oven-dried soil thermal conductivity | (W m$^{-1}$ K$^{-1}$) |
| $\emptyset$ | soil porosity | (%) |
| $\omega$ | annual frequency ($2\pi/365$ or $2\pi/366$ in any leap years) | (-) |
| $\theta, \theta_r, \theta_s$ | actual, residual, and saturated soil water content | (m$^3$ m$^{-3}$) |
| $\rho_b$ | soil bulk density | (g cm$^{-3}$) |
| $b_1, b_2$ | shape factors of soil thermal conductivity curve | (-) |
| $C$ | soil volumetric heat capacity | (J m$^{-3}$ K$^{-1}$) |
| $D_s$ | snow depth | (m) |
| $DR_{eff}$ | effective soil damping ratio | (-) |
| $E, ET_0$ | actual and reference evapotranspiration | (mm) |
| $f_{cl}, f_m, f_{OM}, f_{sa}$ | clay, mineral, organic matter, and sand content in the soil profile | (%) |
| $f_S$ | empirical parameters of snow depth | (m$^{-1}$) |
| $h$ | soil depth | (m) |
| $j$ | day of year | (day) |
| $k_0$ | empirical parameter of soil damping ratio | (-) |
| $k_s$ | soil thermal diffusivity | (m$^2$ s$^{-1}$) |
| $p$ | period of year (365 days or 366 days in any leap year) | days |
| $P$ | precipitation | mm |
| $R_s$ | solar radiation | (MJ m$^{-2}$ d$^{-1}$) |
| $T_a, T_{max}$ | mean and maximum air temperature at 2 m height | (ºC) |
| $T_{env}$ | fictive environmental temperature | (ºC) |
| $T_s$ | bared soil temperature at 0.1 m depth | (ºC) |
| $T_{sfc}$ | surface skin temperature | (ºC) |
| RMSE, MAE | root mean square error and mean absolute error | (ºC) |





**Table A2.** List of datasets used in this study including the data source (Networks), state names
(Coverage States), and specific data variables (Variables). Data sources include the Gridded Soil Survey
Geographic (gSSURGO), the Automated Weather Data Network – High Plains Regional Climate
Center (AWDN), the Oklahoma Mesonet (OK Mesonet), the Soil Climate Analysis Network (SCAN),
and the daily Global History Climatology Network (dGHCN). Weather stations from four states were
located in the U.S. Great Plains including Nebraska (NE), Kansas (KS), Oklahoma (OK), and Texas
(TX). Climate data reports daily maximum ($T_{max}$, ºC) and minimum air temperature ($T_{min}$, ºC) at 2 m
height, relative humidity ($RH$, %), rainfall ($prcp$, mm), solar radiation ($Rs$, MJ m$^{-2}$ day$^{-1}$), wind speed at
2 m ($WS$, m s$^{-1}$), and snow depth ($D_s$, mm). Soil data consists of the daily bare soil temperature at 10 cm
depth ($T_{s,}$ ºC), albedo of soil surface ($\alpha$, -), organic matter content ($f_{OM}$, %), bulk density ($\rho_b$, g cm$^{-3}$),
porosity (Ø, %), sand ($f_{sa}$), silt ($f_{sl}$), and clay ($f_{cl}$) content (%).

| Networks | Coverage States | Variables |
|---|---|---|
| gSSURGO | NE, KS, OK, TX | $\alpha$, $f_{OM}$, $\rho_b$, Ø, $f_{sa}$, $f_{sl}$, and $f_{cl}$ |
| AWDN | NE and KS | $T_{max}$, $T_{min}$, $RH$, $prcp$, $R_s$, $WS$, and $T_s$ |
| OK Mesonet | OK | $T_{max}$, $T_{min}$, $RH$, $prcp$, $R_s$, $WS$, and $T_s$ |
| SCAN | TX | $T_{max}$, $T_{min}$, $RH$, $prcp$, $R_s$, $WS$, and $T_s$ |
| dGHCN | NE, KS, OK, TX | $D_s$ |




**Table A3.** List of model parameters for each weather station in the U.S. Great Plains. The location consists of latitude (Lat) and longitude (Lon). There are 12 parameters in the improved EM model including parameters of air temperature ($\beta$, -); parameters for current day to previous three-day of $T_{env}$: $\alpha_0$ (-), $\alpha_1$ (-), $\alpha_2$ (-), $\alpha_3$ (-), and constant offset $\gamma$ (ºC); annual and semi-annual waves of sine and cosine functions parameters: $\beta_1$, $\beta_2$, $\delta_1$, $\delta_2$ (ºC); parameters for snow depth damping factor ($f_S$, m$^{-1}$) and the soil damping factor ($k_0$, -). The bold font indicates that estimated coefficients are not statistically significant at 95% confidence intervals.

| Location | | | | | | | | | | | | | |
|----------|------|---------|------------|------------|------------|------------|-------|------------|------------|-----------|------------|--------|--------|
| Lat | Lon | $\beta$ | $\alpha_0$ | $\alpha_1$ | $\alpha_2$ | $\alpha_3$ | $\gamma$ | $\beta_1$ | $\delta_1$ | $\beta_2$ | $\delta_2$ | $f_S$ | $k_0$ |
| 26.52 | -98.06 | 0.2 | 0.402 | 0.132 | -0.18 | 0.237 | 7.684 | -0.895 | -2.212 | -0.233 | -0.171 | -0.05 | -0.001 |
| 29.33 | -103.2 | 0.3 | 0.517 | 0.162 | -0.22 | 0.174 | 6.221 | -0.717 | -3.852 | 0.037 | -0.398 | -0.106 | -0.001 |
| 30.27 | -97.74 | 0.8 | 0.247 | 0.191 | 0.017 | 0.1 | 8.416 | -1.373 | -3.454 | 0.03 | -0.269 | -0.079 | -0.001 |
| 31.62 | -102.8 | 0.3 | 0.369 | 0.193 | -0.13 | 0.165 | 6.647 | -1.195 | -4.451 | 0.127 | -0.365 | 0.093 | -0.001 |
| 32.75 | -97 | 0.8 | 0.216 | 0.217 | 0.015 | 0.088 | 9.768 | -1.338 | -2.746 | -0.048 | -0.167 | 0.001 | 0.002 |
| 33.59 | -102.4 | 0.3 | 0.359 | 0.186 | -0.161 | 0.163 | 5.656 | -1.153 | -4.435 | 0.335 | -0.314 | 0.696 | -0.064 |
| 33.63 | -102.8 | 0.1 | 0.421 | 0.087 | -0.175 | 0.228 | 4.153 | -1.366 | -4.637 | 0.201 | -0.077 | -0.27 | -0.047 |
| 33.89 | -97.27 | 0.7 | 0.415 | 0.196 | -0.01 | 0.054 | 4.712 | -0.732 | -4.197 | 0.18 | 0.019 | 1.187 | -0.034 |
| 33.96 | -102.8 | 0.3 | 0.411 | 0.14 | -0.13 | 0.134 | 4.949 | -1.002 | -4.61 | 0.456 | -0.256 | 0.695 | 0.001 |
| 34.03 | -95.54 | 0.8 | 0.535 | 0.136 | -0.003 | 0.058 | 4.076 | -1.164 | -2.726 | 0.306 | 0.065 | -2.045 | -0.049 |
| 34.04 | -96.94 | 0.6 | 0.475 | 0.143 | -0.054 | 0.064 | 5.196 | -1.152 | -4.172 | 0.199 | 0.054 | 2.23 | -0.066 |
| 34.17 | -97.99 | 0.7 | 0.39 | 0.103 | -0.02 | 0.056 | 5.867 | -1.032 | -3.753 | 0.136 | -0.173 | 0.232 | -0.156 |
| 34.19 | -97.59 | 0.8 | 0.407 | 0.099 | 0.009 | 0.043 | 4.471 | -0.809 | -3.197 | 0.085 | -0.091 | -1.257 | -0.194 |
| 34.22 | -95.25 | 0.8 | 0.408 | 0.154 | 0.018 | 0.049 | 5.564 | -1.358 | -3.863 | 0.104 | 0.168 | -4.133 | -0.053 |
| 34.31 | -96 | 0.8 | 0.476 | 0.097 | 0.001 | 0.048 | 5.499 | -1.161 | -3.493 | 0.06 | 0.015 | 0.133 | -0.108 |
| 34.31 | -94.82 | 0.9 | 0.408 | 0.139 | 0.025 | 0.062 | 5.947 | -0.934 | -3.235 | 0.066 | 0.006 | 6.601 | -0.065 |






| Location | | | | | | | | | | | | | |
| --- | --- | --- | --- | --- | --- | --- | --- | --- | --- | --- | --- | --- | --- |
| | | | | | | | Parameters in iEM02 | | | | | | |
| lat | lon | $\beta$ | $\alpha_0$ | $\alpha_1$ | $\alpha_2$ | $\alpha_3$ | $\gamma$ | $\beta_1$ | $\delta_1$ | $\beta_2$ | $\delta_2$ | $f_S$ | $k_0$ |
| 34.57 | -96.95 | 0.5 | 0.451 | 0.142 | -0.083 | 0.091 | 6.331 | -1.225 | -4.095 | 0.016 | 0.075 | -1.584 | -0.079 |
| 34.59 | -99.34 | 0.9 | 0.266 | 0.158 | 0.032 | 0.071 | 6.781 | -1.68 | -4.029 | 0.345 | -0.198 | 3.081 | -0.094 |
| 34.61 | -96.33 | 0.5 | 0.502 | 0.176 | -0.081 | 0.099 | 4.733 | -1.073 | -4.454 | 0.103 | -0.234 | -4.241 | -0.015 |
| 34.66 | -95.33 | 0.9 | 0.466 | 0.165 | 0.021 | 0.06 | 5.079 | -0.917 | -3.484 | 0.13 | 0.183 | -14.03 | -0.029 |
| 34.69 | -99.83 | 0.7 | 0.49 | 0.153 | -0.029 | 0.056 | 5.682 | -1.341 | -4.035 | 0.139 | 0.121 | **-0.068** | -0.048 |
| 34.73 | -98.57 | 0.9 | 0.338 | 0.134 | 0.019 | 0.055 | 4.763 | -1.015 | -3.158 | 0.18 | **0.003** | 1.937 | -0.179 |
| 34.8 | -96.67 | 0.7 | 0.454 | 0.105 | -0.02 | 0.065 | 5.742 | -1.185 | -3.628 | 0.088 | **-0.056** | -1.26 | -0.102 |
| 34.81 | -98.02 | 0.8 | 0.328 | 0.138 | **0.008** | 0.053 | 5.727 | -1.1 | -3.811 | 0.124 | **-0.05** | 0.761 | -0.129 |
| 34.88 | -95.78 | 0.8 | 0.404 | 0.111 | **0.005** | 0.052 | 4.723 | -1.099 | -3.134 | 0.179 | **0.018** | **-0.012** | -0.2 |
| 35.03 | -97.91 | 0.7 | 0.52 | 0.141 | -0.028 | 0.052 | 5.008 | -1.008 | -3.33 | 0.136 | 0.163 | 0.845 | -0.065 |
| 35.19 | -102.1 | 0.6 | 0.239 | 0.139 | -0.014 | 0.088 | 5.715 | -1.807 | -4.526 | 0.267 | -0.208 | -0.129 | -0.131 |
| 35.27 | -97.96 | 0.8 | 0.381 | 0.176 | **0.011** | 0.055 | 5.053 | -1.161 | -3.341 | 0.22 | -0.093 | -0.502 | -0.101 |
| 35.51 | -98.78 | 0.8 | 0.39 | 0.17 | **0.006** | 0.05 | 3.781 | -0.779 | -2.908 | 0.078 | -0.321 | 4.582 | -0.16 |
| 35.55 | -99.73 | 0.5 | 0.533 | 0.124 | -0.09 | 0.116 | 4.372 | -1.198 | -3.711 | 0.291 | **-0.041** | 0.086 | -0.062 |
| 35.58 | -95.91 | 0.7 | 0.391 | 0.152 | **-0.015** | 0.05 | 5.284 | -1.013 | -4.053 | 0.128 | 0.114 | 1.057 | -0.113 |
| 35.59 | -99.27 | 0.8 | 0.467 | 0.136 | **0.011** | 0.056 | 5.461 | -1.244 | -3.594 | 0.182 | 0.143 | 1.237 | -0.042 |
| 35.68 | -94.85 | 0.6 | 0.446 | 0.148 | -0.049 | 0.076 | 4.758 | -1.353 | -4.102 | 0.187 | 0.248 | -3.491 | -0.059 |
| 35.84 | -96 | 0.6 | 0.341 | 0.178 | -0.026 | 0.097 | 6.618 | -1.783 | -4.72 | 0.358 | -0.191 | **-1.872** | -0.023 |
| 35.85 | -97.48 | 0.9 | 0.333 | 0.174 | 0.021 | 0.066 | 5.67 | -1.493 | -3.899 | 0.143 | 0.083 | 0.914 | -0.092 |
| 35.97 | -94.99 | 0.7 | 0.45 | 0.136 | **-0.016** | 0.05 | 5.637 | -1.502 | -3.517 | 0.233 | 0.094 | **0.225** | -0.047 |
| 36 | -97.05 | 0.7 | 0.414 | 0.114 | **-0.007** | 0.038 | 4.93 | -0.909 | -3.849 | 0.154 | **-0.037** | 3.355 | -0.156 |
| 36.03 | -96.5 | 0.7 | 0.385 | 0.149 | **-0.012** | 0.053 | 5.87 | -1.13 | -4.099 | 0.093 | **-0.006** | 2.131 | -0.088 |
| 36.07 | -99.9 | 0.7 | 0.354 | 0.176 | **-0.002** | 0.055 | 5.608 | -1.196 | -4.61 | 0.169 | -0.252 | 1.315 | -0.075 |
| 36.12 | -97.1 | 0.7 | 0.377 | 0.172 | **-0.009** | 0.056 | 5.897 | -1.314 | -4.027 | 0.162 | **0.031** | 2.05 | -0.062 |
| 36.26 | -98.5 | 0.7 | 0.429 | 0.15 | -0.021 | 0.055 | 4.908 | -1.27 | -4.183 | 0.25 | 0.265 | **0.169** | -0.067 |
| 36.41 | -97.69 | 0.6 | 0.397 | 0.175 | -0.038 | 0.079 | 5.029 | -1.188 | -4.643 | 0.089 | -0.276 | 1.137 | -0.062 |
| 36.42 | -96.04 | 0.9 | 0.36 | 0.144 | 0.017 | 0.048 | 5.533 | -1.159 | -3.882 | 0.118 | 0.069 | 3.667 | -0.104 |
| 36.52 | -96.34 | 0.7 | 0.308 | 0.157 | **-0.005** | 0.063 | 5.684 | -1.494 | -4.43 | 0.29 | 0.149 | 0.563 | -0.13 |
| 36.6 | -101.6 | 0.7 | 0.453 | 0.173 | **-0.015** | 0.078 | 5.052 | -1.21 | -3.919 | 0.261 | **-0.057** | 1.796 | -0.03 |
| 36.63 | -96.81 | 0.6 | 0.545 | 0.151 | -0.08 | 0.067 | 4.993 | -0.829 | -3.553 | 0.061 | **0.031** | **0.66** | -0.061 |
| 36.69 | -102.5 | 0.7 | 0.223 | 0.16 | **0.008** | 0.057 | 5.711 | -1.795 | -5.263 | 0.235 | -0.155 | 1.458 | -0.125 |
| 36.75 | -98.36 | 0.5 | 0.474 | 0.152 | -0.083 | 0.082 | 4.968 | -1.198 | -4.11 | 0.191 | **-0.011** | 3.696 | -0.072 |
| 36.75 | -97.25 | 0.7 | 0.383 | 0.167 | -0.02 | 0.056 | 5.197 | -1.086 | -4.062 | 0.005 | **0.001** | 4.13 | -0.083 |
| 36.83 | -99.64 | 0.7 | 0.354 | 0.189 | **-0.004** | 0.074 | 5.242 | -1.28 | -4.03 | 0.204 | **0.047** | **0.367** | -0.054 |
| 36.84 | -96.43 | 0.6 | 0.382 | 0.209 | -0.028 | 0.066 | 5.279 | -1.386 | -4.9 | 0.418 | -0.258 | -7.259 | -0.014 |
| 36.9 | -96.91 | 0.6 | 0.374 | 0.16 | -0.031 | 0.06 | 5.007 | -1.266 | -4.123 | 0.152 | **0.021** | 4.473 | -0.103 |
| 36.91 | -95.89 | 0.6 | 0.415 | 0.169 | -0.045 | 0.053 | 5.844 | -1.22 | -4.352 | 0.235 | -0.255 | 6.361 | -0.063 |
| 37.37 | -95.3 | 0.7 | 0.373 | 0.173 | **-0.016** | 0.076 | 4.535 | -1.64 | -4.396 | **0.047** | **-0.009** | **-0.77** | -0.021 |



| Location | | | | | | | | | | | | | |
|---|---|---|---|---|---|---|---|---|---|---|---|---|---|
| lat | lon | $\beta$ | $\alpha_0$ | $\alpha_1$ | $\alpha_2$ | $\alpha_3$ | $\gamma$ | $\beta_1$ | $\delta_1$ | $\beta_2$ | $\delta_2$ | $f_S$ | $k_0$ |
| 37.98 | -100.8 | 0.7 | 0.346 | 0.185 | **0.001** | 0.077 | 4.153 | -1.319 | -4.886 | 0.084 | -0.441 | 1.602 | -0.074 |
| 38.45 | -101.8 | 0.4 | 0.297 | 0.142 | -0.073 | 0.124 | 5.091 | -2.004 | -6.426 | 0.18 | -0.585 | 1.573 | -0.038 |
| 38.53 | -95.25 | 0.6 | 0.46 | 0.176 | -0.053 | 0.071 | 3.971 | -1.302 | -3.761 | 0.128 | -0.075 | 1.529 | -0.054 |
| 39.07 | -95.78 | 0.6 | 0.387 | 0.162 | -0.033 | 0.089 | 4.229 | -1.715 | -4.991 | **0.019** | 0.126 | **0.279** | -0.048 |
| 39.2 | -96.6 | 0.5 | 0.4 | 0.163 | -0.077 | 0.107 | 4.461 | -1.724 | -4.951 | **-0.021** | 0.081 | 1.77 | -0.025 |
| 39.38 | -101.1 | 1 | 0.252 | 0.188 | 0.055 | 0.075 | 4.641 | -1.501 | -5.624 | 0.062 | -0.369 | 1.507 | -0.078 |
| 39.82 | -97.85 | 0.8 | 0.437 | 0.182 | **0.008** | 0.058 | 3.385 | -1.482 | -4.33 | **-0.004** | -0.109 | **-0.429** | -0.068 |
| 40.08 | -98.28 | 0.5 | 0.44 | 0.151 | -0.076 | 0.076 | 4.164 | -1.339 | -4.958 | **-0.017** | -0.093 | 7.636 | -0.046 |
| 40.3 | -96.93 | 0.7 | 0.381 | 0.205 | **-0.014** | 0.072 | 3.293 | -1.615 | -4.407 | 0.204 | 0.088 | **0.342** | -0.064 |
| 40.32 | -99.38 | 0.6 | 0.319 | 0.199 | -0.027 | 0.055 | 4.414 | -1.593 | -5.523 | 0.25 | **-0.063** | 6.898 | -0.1 |
| 40.4 | -101.7 | 0.6 | 0.364 | 0.145 | -0.031 | 0.079 | 3.559 | -1.151 | -5.266 | 0.402 | -0.162 | 8.794 | -0.12 |
| 40.5 | -99.37 | 0.6 | 0.337 | 0.202 | -0.029 | 0.08 | 4.765 | -1.676 | -5.194 | 0.307 | 0.153 | 2.688 | -0.029 |
| 40.52 | -99.05 | 0.6 | 0.379 | 0.172 | -0.031 | 0.068 | 3.676 | -1.45 | -4.852 | 0.281 | **-0.026** | 5.941 | -0.086 |
| 40.57 | -99.7 | 0.5 | 0.329 | 0.18 | -0.051 | 0.085 | 3.856 | -1.901 | -5.549 | 0.302 | 0.214 | 9.566 | -0.085 |
| 40.57 | -98.15 | 0.8 | 0.28 | 0.158 | **0.013** | 0.056 | 4.244 | -1.671 | -4.596 | 0.205 | 0.072 | **0.708** | -0.245 |
| 40.63 | -100.5 | 0.7 | 0.36 | 0.199 | **-0.015** | 0.064 | 3.888 | -1.484 | -5.794 | 0.089 | -0.136 | 3.376 | -0.054 |
| 40.72 | -99.02 | 0.6 | 0.406 | 0.195 | -0.034 | 0.076 | 3.572 | -1.456 | -5.1 | 0.205 | **0.043** | **0.756** | -0.032 |
| 40.75 | -98.77 | 0.5 | 0.437 | 0.158 | -0.075 | 0.081 | 3.778 | -1.502 | -5.411 | 0.35 | 0.097 | 2.179 | -0.078 |
| 40.82 | -96.67 | 0.6 | 0.384 | 0.193 | -0.048 | 0.066 | 4.302 | -1.619 | -4.854 | 0.111 | 0.079 | 3.63 | -0.102 |
| 40.85 | -96.62 | 0.2 | 0.588 | 0.032 | -0.209 | 0.173 | 3.744 | -1.616 | -4.753 | 0.141 | 0.275 | 11.447 | -0.048 |
| 40.86 | -98.47 | 0.5 | 0.521 | 0.151 | -0.089 | 0.074 | 2.731 | -1.53 | -4.791 | 0.114 | 0.309 | 7.99 | -0.067 |
| 41.15 | -96.5 | 0.7 | 0.354 | 0.169 | **-0.011** | 0.065 | 4.615 | -1.819 | -4.679 | 0.124 | **0.05** | 4.403 | -0.07 |
| 41.15 | -96.42 | 0.6 | 0.42 | 0.172 | -0.055 | 0.071 | 3.925 | -1.883 | -5.022 | 0.105 | 0.102 | 3.669 | -0.053 |
| 41.22 | -103 | 0.7 | 0.323 | 0.188 | **-0.001** | 0.054 | 3.392 | -1.118 | -5.154 | 0.263 | -0.119 | 10.875 | -0.072 |
| 41.4 | -97.53 | 0.5 | 0.489 | 0.131 | -0.082 | 0.082 | 3.624 | -1.5 | -4.763 | 0.389 | 0.074 | 5.878 | -0.057 |
| 41.62 | -98.95 | 0.6 | 0.403 | 0.164 | -0.039 | 0.077 | 3.52 | -1.649 | -5.573 | 0.136 | 0.093 | 2.696 | -0.074 |
| 41.85 | -96.75 | 0.7 | 0.336 | 0.201 | -0.025 | 0.076 | 4.125 | -1.82 | -5.053 | 0.296 | 0.099 | 11.111 | -0.057 |
| 41.88 | -103.7 | 0.7 | 0.346 | 0.2 | **-0.007** | 0.07 | 3.58 | -1.41 | -5.435 | 0.44 | -0.079 | 4.335 | -0.061 |
| 41.9 | -100.2 | 0.3 | 0.548 | 0.125 | -0.195 | 0.144 | 2.915 | -1.653 | -4.817 | 0.187 | 0.146 | 1.078 | -0.043 |
| 41.93 | -98.2 | 0.5 | 0.417 | 0.159 | -0.073 | 0.074 | 3.353 | -1.279 | -4.421 | 0.243 | -0.076 | 10.447 | -0.078 |
| 42.47 | -98.77 | 0.5 | 0.509 | 0.13 | -0.104 | 0.091 | 3.542 | -1.505 | -4.489 | 0.129 | 0.231 | 2.505 | -0.056 |
| 42.57 | -99.83 | 0.3 | 0.54 | 0.074 | -0.182 | 0.156 | 3.049 | -1.454 | -5.465 | 0.202 | 0.286 | 1.976 | -0.077 |
| 42.75 | -102.2 | 0.7 | 0.43 | 0.144 | -0.024 | 0.068 | 2.82 | -1.142 | -5.151 | 0.141 | 0.249 | 3.492 | -0.089 |





**Code availability**

MATLAB code is available upon request.


**Data availability**

Data used in this study are available by the links above (material and methods).

**Author contribution**

HD and XL designed the experiments, conducted simulations, analyzed the data, and wrote the manuscript. NW helped data analysis, result interpretation, and discussion. MBK and GFS provided suggestions and discussion, wrote and revised the manuscript.

**Competing interests**

The authors declare that they have no conflict of interest.

**Acknowledgements**

This study was supported in part by the U.S. Department of Agriculture (ARS grant no. 58-3090-5-009 and NIFA grant no. 2016-68007- 25066) and the Kansas Crop Improvement Association, the U.S.
Department of Agriculture National Institute of Food and Agriculture, Hatch project 1018005. The contribution number of this manuscript is 21-250-J. We appreciated Dr. Gerard Kluitenberg and Dr. Jesse Tack at Kansas State University for providing helpful suggestions to improve the quality of paper. We thank Dallas Staley for her outstanding contribution in editing and finalizing the paper. Her work continues to be at the highest professional level.




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





**Figure Captions (9 Figures)**

**Figure 1:** Specific soil textures (**a**) and soil bulk density (**b**) at 87 weather stations in the U.S. winter
wheat belt including the states of Nebraska (NE), Kansas (KS), Oklahoma (OK), and part of Texas (TX)
in the U.S. Great Plains.

**Figure 2:** Effects of soil moisture on (**a**) thermal conductivity ($\lambda$) and (**b**) soil thermal diffusivity ($k$)
obtained by Eqs. (7-11).

**Figure 3**: Percentage increments of soil temperature modeling improvement in iEM02 as determined by
RMSE changes $[-\frac{100(RMSE_{improved} - RMSE_{original})}{RMSE_{original}}]$ (**a**) after introducing air temperature of $T_{a, j-3}$, (**b**)
after substituting air temperature $T_a$ by fictive environmental temperature ($T_{env}$), and (**c**) after integrating
the impacts of soil thermal diffusivity and snow cover. The colorbar was coded by the improved
percentage of iEM02 against the EM02 model.

**Figure 4:** Spatial variations of the improved empirical model (iEM) coefficients: (**a-d**) for $\alpha_0$, $\alpha_1$, $\alpha_2$,
and $\alpha_3$, (**e-h**) for $\beta_1$, $\delta_1$, $\beta_2$, and $\delta_2$, (**i**) snow damping ratio ($f_s$) and (**j**) soil damping ratio coefficients
($k_0$). The colorbar defines the values of the model's coefficients.

**Figure 5:** One-to-one plots of absolute mean errors between the complete model and reduced model in
the improved empirical model (iEM02): (**a-d**) with vs. without $\alpha_4$ in Nebraska (NE), Kansas (KS),
Oklahoma (OK), and Texas (TX), respectively, (**e-h**) with vs. without $\beta_1$ in Nebraska (NE), Kansas
(KS), Oklahoma (OK), and Texas (TX), respectively; (**i-l**) with vs. without $\delta_1$ in NE, KS, OK, and TX,
respectively. RMSE_reduced and RMSE_complete refer to root mean square error for reduced and complete
models, respectively. The colorbar indicates the number of observed data points.

**Figure 6:** Spatial distribution of mean absolute error (MAE) (a, c) and RMSE (b, d) for an empirical
model (EM02, a, b), and improved modelEM02 (iEM02, c, d). The colorbar defines values of MAE
(°C) and RMSE (°C).

**Figure 7:** Seasonal comparison between estimated and observed soil temperatures: (**a-d**) the empirical
model (EM), and (**e-h**) the improved empirical model (iEM02). RMSE was calculated as the root mean
square error between estimated and observed soil temperature. "N" refers to the sample size and the
gray line represents the 1:1 line. The colorbar describes the number of data points.

**Figure 8:** Comparison between observed (grey line), complete model (EM02, green line) and improved
model (iEM02, blue line) daily soil temperature in western (>100°W), central (between 97° and 100°
W), and eastern (<97°W) Nebraska (**a-c**) and Kansas (**d-f**) during the winter wheat growing seasons
from 2015 to 2019. RMSE is the root mean square error (°C). Shaded areas indicate winter season (Dec-
Feb).

**Figure 9:** The same as Fig. 9 but for western, central, and eastern Oklahoma (**a-c**) and Texas (**d-f**).



**Figure 1.**

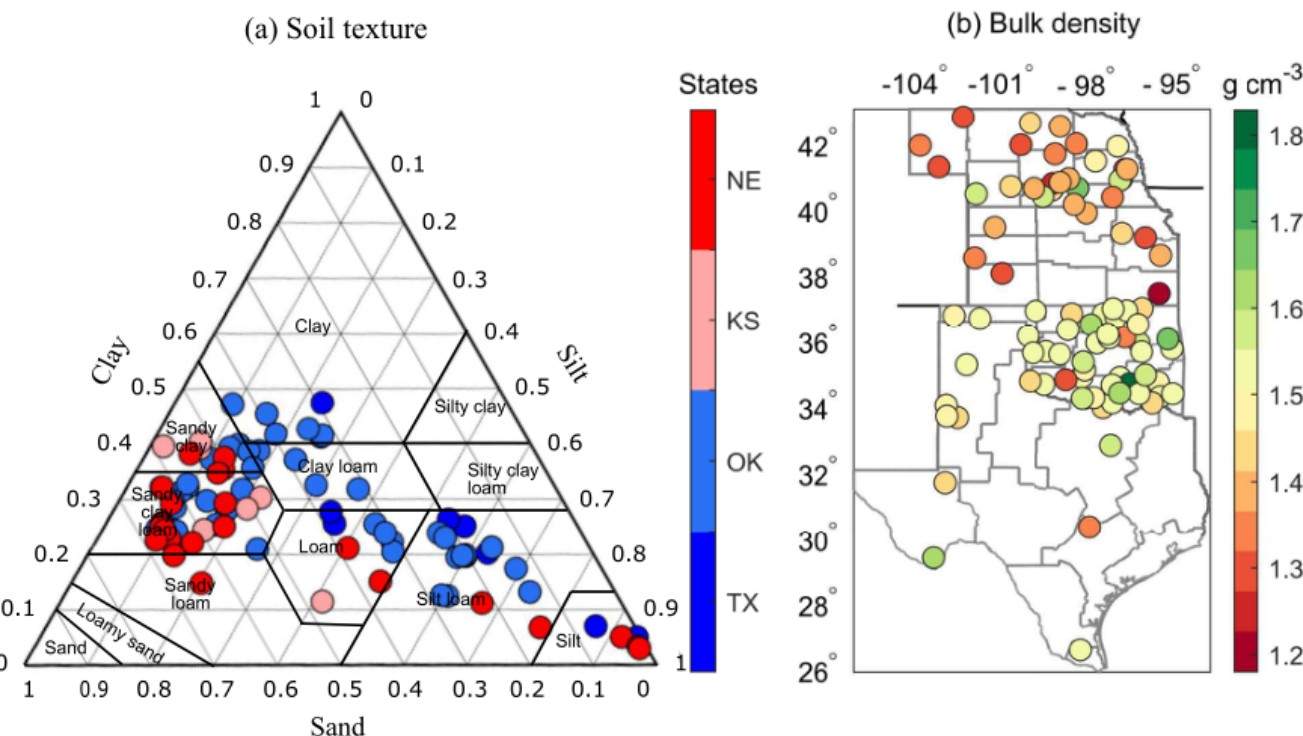

**Figure 1:** Specific soil textures (**a**) and soil bulk density (**b**) at 87 weather stations in the U.S. winter wheat belt including the states of Nebraska (NE), Kansas (KS), Oklahoma (OK), and part of Texas (TX) in the U.S. Great Plains.



**Figure 2.**

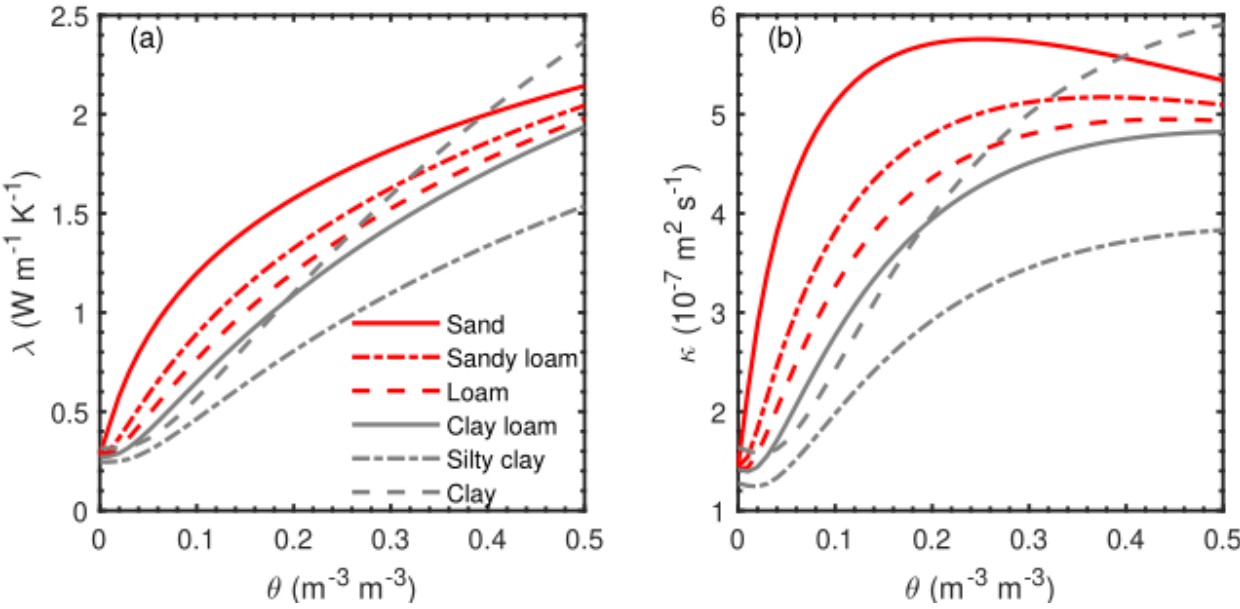

**Figure 2:** Effects of soil moisture on (**a**) thermal conductivity ($\lambda$) and (**b**) soil thermal diffusivity ($k$)
obtained by Eqs. (7-11).


**Figure 3.**

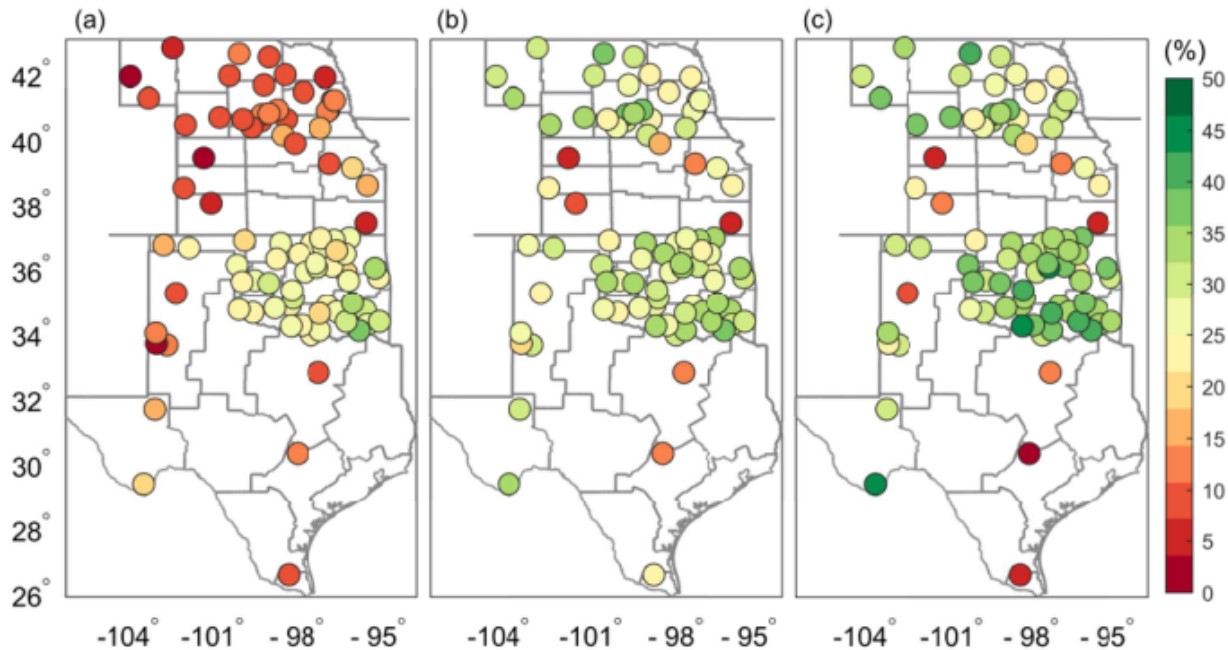

**Figure 3**: Percentage increments of soil temperature modeling improvement in iEM02 as determined by

RMSE changes $[-\frac{100(RMSE_{improved} - RMSE_{original})}{RMSE_{original}}]$ (**a**) after introducing air temperature of $T_{a,\,j-3}$, (**b**)

after substituting air temperature $T_a$ by fictive environmental temperature ($T_{env}$), and (**c**) after integrating the impacts of soil thermal diffusivity and snow cover. The colorbar was coded by the improved percentage of iEM02 against the EM02 model.


**Figure 4.**

**Figure 4:** Spatial variations of the improved empirical model (iEM) coefficients: **(a-d)** for $\alpha_0$, $\alpha_1$, $\alpha_2$, and $\alpha_3$, **(e-h)** for $\beta_1$, $\delta_1$, $\beta_2$, and $\delta_2$, **(i)** snow damping ratio ($f_s$) and **(j)** soil damping ratio coefficients ($k_0$). The colorbar defines the values of the model's coefficients.



**Figure 5.**



**Figure 5:** One-to-one plots of absolute mean errors between the complete model and reduced model in the improved empirical model (iEM02): (**a-d**) with vs. without $\alpha_4$ in Nebraska (NE), Kansas (KS), Oklahoma (OK), and Texas (TX), respectively, (**e-h**) with vs. without $\beta_1$ in Nebraska (NE), Kansas (KS), Oklahoma (OK), and Texas (TX), respectively; (**i-l**) with vs. without $\delta_1$ in NE, KS, OK, and TX, respectively. $RMSE_{reduced}$ and $RMSE_{complete}$ refer to root mean square error for reduced and complete models, respectively. The colorbar indicates the number of observed data points.


**Figure 6.**

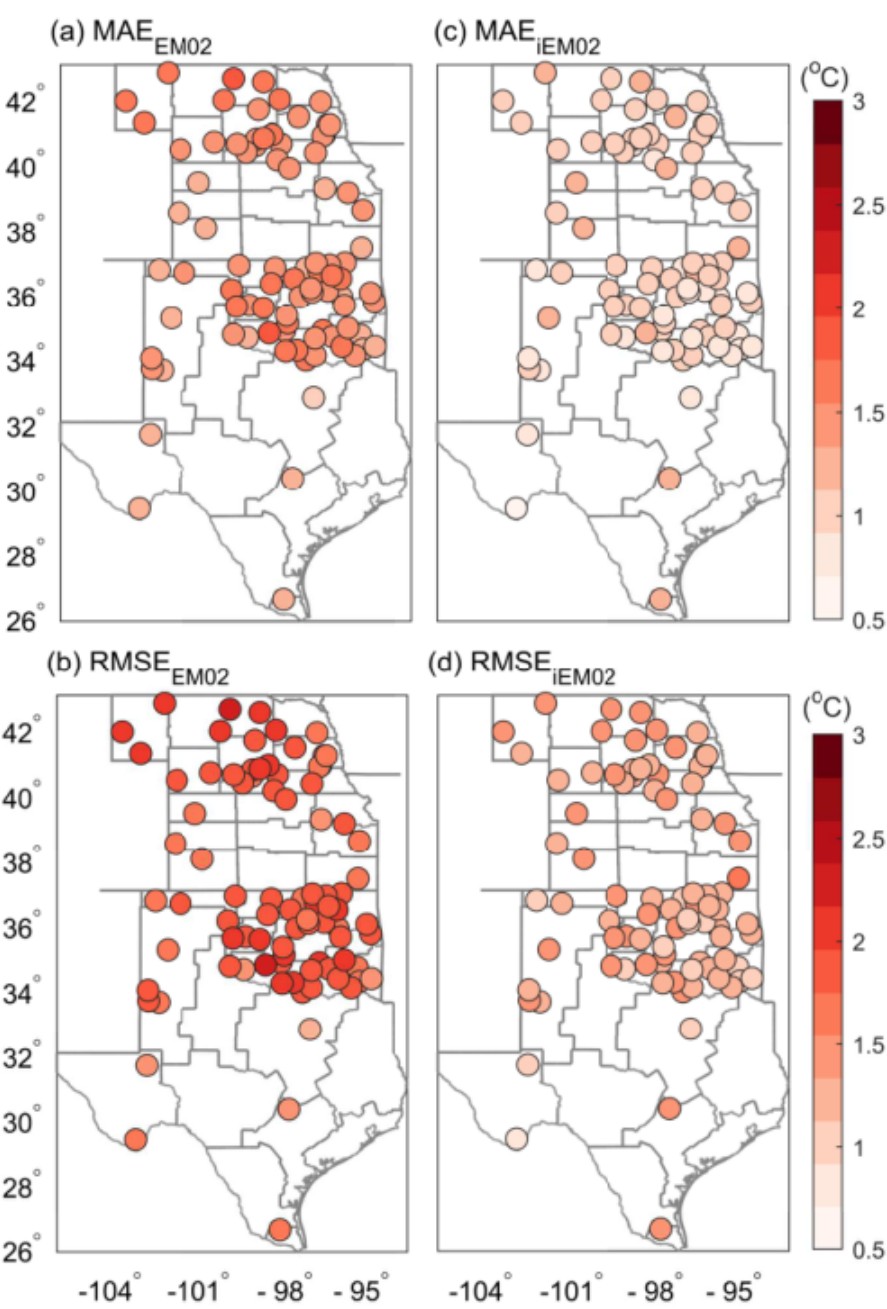


**Figure 6:** Spatial distribution of mean absolute error (MAE) (a, c) and RMSE (b, d) for an empirical model (EM02, a, b), and improved modelEM02 (iEM02, c, d). The colorbar defines values of MAE (ºC) and RMSE (ºC).


**Figure 7.**

![Figure 7 scatter density plots comparing estimated and observed soil temperatures across seasons for EM02 and iEM02 models]

**Figure 7:** Seasonal comparison between estimated and observed soil temperatures: (**a-d**) the empirical model (EM), and (**e-h**) the improved empirical model (iEM02). RMSE was calculated as the root mean square error between estimated and observed soil temperature. "N" refers to the sample size and the gray line represents the 1:1 line. The colorbar describes the number of data points.



**Figure 8.**

**Figure 8:** Comparison between observed (grey line), complete model (EM02, green line) and improved model (iEM02, blue line) daily soil temperature in western (>100ºW), central (between 97º and 100º W), and eastern (<97ºW) Nebraska (**a-c**) and Kansas (**d-f**) during the winter wheat growing seasons from 2015 to 2019. RMSE is the root mean square error (ºC). Shaded areas indicate winter season (Dec-

Feb).





**Figure 9.**

**Figure 9:** The same as Fig. 9 but for western, central, and eastern Oklahoma (**a-c**) and Texas (**d-f**).