# Peer review of "Daily soil temperature modeling improved by integrating observed snow cover and estimated soil moisture in the U.S. Great Plains"

_Hydrology and Earth System Sciences, 2021_

## Author Comment (AC1)

**Point-by-point Responses to two Referees for # hess-2021-164**

**Referee 1 for # hess-2021-164**

The authors have used regular fonts for the Referee's comments and blue fonts for our responses and red fonts with quotation marks to show the revised text.

This manuscript focuses on a very important subject --- how to improve the model in better predicting soil temperature at the soil surface layer. The topic is well within the scientific scopes of HESS because soil temperature data are critical in many research areas, such as meteorological, hydrological and ecosystem modeling, agricultural, soil and plant studies, and so on. The paper itself is generally well written and structured. Although the proposed method and model were developed for and tested in the region of U.S. Great Plains only, they can be easily applied to other regions in the world. The conclusions were sound and based on their data and figures. I have made some comments that may help authors in improving their manuscript. I strongly recommend that this manuscript should be accepted and published on HESS after the following comments are addressed.

**Response:** Thank you for your review and insight, which improved our paper. We responded to all of your comments.

Major comments:
1. On page 7 authors stated "the current empirical model was selected because it was more accurate than the process-based model". Then on page 14 (lines 261-262), they chose not to show the evidence to support the above statement. Theoretically, the energy balance models are physically sound and should predict the soil temperature more accurately if they are used properly. If no evidence or explanation is provided in this manuscript, it is difficult to convince readers that the statement on page 7 is true. If the paper length is a concern, authors at least should offer some explanations in the discussion session why the process-based models (energy balance models) failed in their studied region, or in other words, why their proposed model was a better choice than the energy balance model in this region.

**Response:** We appreciate your insights and suggested revisions. The authors initially studied both an empirical model and a surface energy-balance model (EBM, Chalhoub et al., 2017) for all selected stations in this study. Fig. R1.1 below presents the modeling results for both EBM and EM02 when compared to our improved model (iEM02). When we consider the paper's length and its focus, we decided not to include the energy balance model. We described the reason why we excluded the EBM modeling results in our original manuscript (lines 59-62).

[Figure]

**Figure R1.1:** Spatial distribution of mean absolute error (MAE) (a, c, and e) and RMSE (b, d, and f) for the energy balance model (Chalhoub et al., 2017) (EBM, a, b), the empirical model (EM02, c, d), and an improved empirical model (iEM02, e, f), respectively. The colorbar defines values of MAE (°C) and RMSE (°C).

2. Section 2.2.2: authors may want to provide more rationales or explanations in designing their improved model. They introduced several new parameters to the model to represent the physical processes related to soil temperature. Some of those can be easily understood, such as soil thermal conductivity, thermal diffusivity and snow depth while the reasons for others were not very obvious. For example, why was it necessary to create a fictive environmental temperature (Tenv, °C) in replacement of air temperature (Ta)? What was the reason to include Tenv from an extra prior day (j-3)? Their results in Figure 4 did not show that this extra inclusion led to a significant improvement of model outputs.

**Response:** Thanks for these useful suggestions. We agree. We added and modified sentences in our revision to clarify $T_{env}$ and $T_{a, j-3}$. The modeling improvement by using the fictive environmental temperature ($T_{env}$) was significant in northern areas of NE and KS (See Fig. 3b). For example, an approximately 15% improvement in simulated soil temperature was achieved in NE and KS when $T_a$ was replaced by $T_{env}$. We have modified the text to explain the importance of $T_{env}$ in the model, as follows:

"where a fictive environmental temperature ($T_{env}$) is assumed to be the weighted mean of air temperature ($T_a$) at 2 m and surface temperature ($T_{sfc}$). β is a partitioning coefficient, which defines the relative weight of the air temperature. This weighted fictive temperature will help weigh surface cooling and heating processes due to radiative and convective process (Dolschak et al., 2015)."

The temperature $T_{env, j-3}$ is added because soil texture changes across our study area and soil temperature usually has a longer memory.

3. Section 2.3: authors evaluated the model by quantifying the errors (RMSE and MAE) or the deviations of magnitude between the model outputs and observations. Another important characteristic of daily mean soil temperature is its seasonal cycle. This can also be an important metric for model validation (whether model outputs are in phase with the observations). One way of doing this can be to test the correlation (or lagged correlation) between the model outputs and observations.

**Response:** Thank you for your suggestions. We analyzed the linear correlation coefficient between predicted and observed soil temperature (Fig. R1.2). The linear correlation is less robust than the RMSE metric in this application.

[Figure]

**Figure R1.2:** Spatial distribution of the correlation coefficient between predicted and observed soil temperature an empirical model (EM02, a) and the improved modelEM02 (iEM02, b). The colorbar defines the correlation coefficient (-).

Minor comments:
1. Line 62 on page 4: 'Kutikoff et al., 2021' and 'Dhungel et al., 2021' were not found in the reference list.

**Response:** Thank you. We have added these two references.

"Kutikoff, S., Lin, X., Evett, S. R., Gowda, P., Brauer, D., Moorhead, J., Marek, G., Colaizzi, P., Aiken, R., Xu, L. K., and Owensby, C. Water vapor density and turbulent fluxes from three generations of infrared gas analyzers. Atmospheric Measurement Techniques, 14(2), 1253-1266, 2021."

"Dhungel, R., Aiken, R., Evett, S. R., Colaizzi, P. D., Marek, G., Moorhead, J. E., Baumhardt, R. L., Brauer, D., Kutikoff, S., and Lin, X. Energy imbalance and evapotranspiration hysteresis under an advective environment: Evidence from lysimeter, eddy covariance, and energy balance modelling. Geophysical Research Letters, 2021."

2. Lines 112-117 on pages 6 and 7: the writing in this part can be confusing and should be rewritten in the following format --- 'In this study, three surface climate datasets were obtained: (1) …; (2)…; and (3)…' or 'In this study, three surface climate datasets were obtained. The first one … . The second one … . The third one …'.

**Response:** Thank you for the suggestion. We have adjusted the sentences in our revision, as shown below:

"In this study, three surface climate datasets were obtained from: (1) Automated Weather Data Networks (AWDN) (https://hprcc.unl.edu/awdn/), commissioned in the 1980s for Nebraska and Kansas; (2) The Oklahoma Mesonet, which is a daily climate data source for Oklahoma and, which started in the 1990s (http://www.mesonet.org/); and (3) the Soil and Climate Analysis Network, which gives daily climate observations (https://www.wcc.nrcs.usda.gov/scan/) that we selected for Texas due to limited quality data available in its automated weather station network."

3. Is it more proper to refer to Figure 2 somewhere between line# 167 and 177 (pages 9 and 10) instead of line# 148 on page 8?

**Response:** Agreed. We added this.

4. Lines 210-213 on page 11: How can soil types at different locations explain the different levels of modeling improvement by including $T_{a, j-3}$? Please be more specific.

**Response:** Thank you for your careful review. Thermal conductivity of clay and silt soil is lower than that of the sand soil. In our manuscript, we described the importance of soil types in Lines 212-213. Here we slightly modified them as below:

"The soil types in OK are more clay and silt compared to NE and KS (Fig. 1). However, the improvement by using the fictive environmental temperature was significant in northern areas of NE and KS (sandy soil) but not in the southern area of OK and part of TX (clay and silt soil) (Fig. 3b)."

5. Lines 240-251 on page 13 and Figure 5: first, you stated 'the $\alpha_0$ term was removed (Fig.5, a-d)' in the text but in caption of Figure 5 you stated '(a-d) with vs. without $\alpha_4$'. This is inconsistent ($\alpha_0$

vs. α4). Second, in Figure 5 shown on the x- and y-axes were simply the difference (absolute error) between the simulated and the observed data. However, in both the text and figure caption you used phrases RMSE and absolute mean errors (neither is a correct description of Figure 5). Please correct them thoroughly.

**Response:** Thank you for your careful review. First, we replaced $\alpha_4$ in the caption by $\alpha_0$. Second, the RMSE was calculated individually in completed and reduced models. We simply displayed this RMSE ratio in Figure 5 to contain both absolute error and RMSE. To clarify, we modified the caption for Figure 5 in our revision as follow:

"Figure 5: One-to-one plots of absolute mean errors between the complete model (EM02) and reduced model where one independent variable term was removed in the improved empirical model (iEM02): (**a-d**) with vs. without $\alpha_0$ in Nebraska (NE), Kansas (KS), Oklahoma (OK), and Texas (TX), respectively, (**e-h**) with vs. without $b_1$ in Nebraska (NE), Kansas (KS), Oklahoma (OK), and Texas (TX), respectively; (**i-l**) with vs. without $d_1$ in NE, KS, OK, and TX, respectively. The ratio of the root mean square error (RMSE) shown includes the reduced RMSE and RMSE complete that refer to root mean square error for reduced and complete models, respectively. The colorbar indicates the number of observed data points."

6. Lines 242-243 on page 13: 'indicating a 20% drop in RMSE' --- did you mean 'a 20% raise'?

**Response:** Thank you for your review. Yes, you are correct. It indicates a 20% increase. We replaced "drop" with "raise" in our revision.

7. Line 268 on page 14: "Nebraska and Oklahoma Similar results" --- insert a period (.) between "Nebraska and Oklahoma" and "Similar results"

**Response:** Done.

8. Figures 8 and 9: with the current settings in these two figures, those lines (EM02, iEM02 and Obs) almost overlaid on each other and the difference between them was hardly identified. This reduced the values of these plots. Consider re-doing them or converting them into tables.

**Response:** To compare modeling performance clearly, we calculated the RMSE for three sub-growing seasons: October-November, December-January-February, and March-April-May-June. The values are shown in Table R1.1. We added these RMSE values inside the revised Figures 8 and 9. We revised the Figure 8's caption, as below:

"**Figure 8:** Daily soil temperature comparison between observed (grey line), complete model (EM02, green line), and improved model (iEM02, blue line) in western (>100°W), central (between 97° and 100°W), and eastern (<97°W) Nebraska (a-c) and Kansas (d-f) during the winter wheat growing seasons from 2015 to 2019. RMSE is the root mean square error (°C). Three values in brackets refer to the RMSE for the periods of October-November, December-January-February, and March-April-May-June. Shaded areas indicate the winter season (December-February)."

**Table R1.1**. RMSE values we added in revised Figures 8 and 9.

| Regions | Model performance | |
| --- | --- | --- |
| | **EM02** | **iEM02** |
| NE_west | RMSE = [0.9 1.1 1.1] | RMSE = [0.5 0.7 0.8] |
| NE_central | RMSE = [1.0 0.9 0.9] | RMSE = [0.6 0.6 0.5] |
| NE_east | RMSE = [1.1 1.1 1.1] | RMSE = [0.7 0.7 0.6] |
| KS_west | RMSE = [1.5 1.2 1.0] | RMSE = [1.1 0.9 0.7] |
| KS_central | RMSE = [1.6 1.8 2.1] | RMSE = [1.4 1.5 1.3] |
| KS_east | RMSE = [1.6 1.2 0.8] | RMSE = [1.2 1.0 0.5] |
| OK_west | RMSE = [1.2 1.3 1.4] | RMSE = [0.8 0.8 0.9] |
| OK_central | RMSE = [1.2 0.8 0.7] | RMSE = [0.7 0.5 0.4] |
| OK_east | RMSE = [1.1 0.7 0.6] | RMSE = [0.6 0.4 0.4] |
| TX_west | RMSE = [0.8 0.7 0.6] | RMSE = [0.5 0.6 0.5] |
| TX_central | RMSE = [0.7 1.1 1.2] | RMSE = [0.8 1.0 0.9] |
| TX_east | RMSE = [1.0 1.4 1.6] | RMSE = [0.9 1.4 1.3] |

**--- The END of point-by-point response for referee #1**

---

## Author Comment (AC2)

**Point-by-point Responses to two Referees for # hess-2021-164**

The authors have used regular fonts for the Referee's comments and blue fonts for our responses and red fonts with quotation marks to show the revised text.

This manuscript seeks to improve an empirical 10 cm bare soil temperature prediction for the Great Plains by incorporating snow cover, soil moisture, and additional previous temperature data. Data from are validated against a multi-state mesonet and show a reduction in root mean squared error. The importance of knowing soil temperature data for hydrologic and agricultural applications is quite clear, the rationale for an empirical approach very understandable, and the key parameters that increase thermal mass (increased soil moisture and cover) are rational for model improvement. The topic is of high relevance for readers of Hydrology and Earth System Sciences.

**Response:** Thank you for your review and insights, which improved our paper. We responded to all of your comments.

Major comments:
1. While I think this manuscript may be a useful contribution to the literature, I have two major comments that I feel need addressing. One is on the input data to the model. The model seeks to predict soil temperature, but it needs soil moisture as an input. Soil moisture seems to be at least as difficult to measure, if not more so, than soil temperature, so the practical utility of this specific model seems suspect. It would have been much more useful if a satellite-based soil moisture/snow cover product such as those available from SMAP or Sentinel (Das et al., 2019) were used as inputs. Similarly, the soil texture product used in this study is at much coarser resolution than products such as POLARIS (Chaney et al., 2016), which are available at 30m resolution. I really think this analysis would be much stronger if these products were used. At the very least, I think more explanation and discussion is needed around this.

References:
Chaney, N. W., Wood, E. F., McBratney, A. B., Hempel, J. W., Nauman, T. W., Brungard, C. W., and Odgers, N. P.: POLARIS: A 30-meter probabilistic soil series map of the contiguous United States, Geoderma, 274, 54–67, https://doi.org/10.1016/j.geoderma.2016.03.025, 2016.

Das, N. N., Entekhabi, D., Dunbar, R. S., Chaubell, M. J., Colliander, A., Yueh, S., Jagdhuber, T., Chen, F., Crow, W., O'Neill, P. E., Walker, J. P., Berg, A., Bosch, D. D., Caldwell, T., Cosh, M. H., Collins, C. H., Lopez-Baeza, E., and Thibeault, M.: The SMAP and Copernicus Sentinel 1A/B microwave active-passive high resolution surface soil moisture product, Remote Sensing of Environment, 233, 111380, https://doi.org/10.1016/j.rse.2019.111380, 2019.

**Response:** Thank you for your insight. We agree that soil moisture and snow cover detected by satellite would be useful if they are available. Yes, the soil moisture modeling is much more challenging than soil temperature modeling due to soil moisture transport mechanisms and its high heterogeneity. We used soil moisture estimated from a simple empirical approach based on reference evapotranspiration, precipitation, and surface water balance. Yes, strictly, this estimate

is not accurate in an absolute sense but it does help for improving soil temperature modeling as a secondary input in our model. We attempted to use the SMAP Level-3, 9 km soil moisture product for 2019. We did spline-interpolation for each station from the 9 km grids. It turns out that when using the SMAP data for soil moisture the modeled soil temperature had a 1.5°C RMSE on average and 26% of stations had larger than 1.6°C RMSE. We then realized that the simple estimated soil moisture for each station performed better than the results by using SMAP. We believe part of the reason for this result was because the 9 km soil moisture resolution was too coarse for our purpose. Figure R1.3 shows the result. However, we also believe that these products (if we could assimilate SMAP and other high spatial and temporal resolution satellites together in near future) would certainly be helpful for the daily soil temperature modeling.

[Figure]

**Figure R1.3:** Spatial distribution of mean absolute error (MAE) and root mean square error (RMSE) for an improved empirical model (iEM02, a and b) using the SMAP soil moisture product for 2019 year as an example. The colorbar defines values of MAE (°C) and RMSE (°C).

For the second question associated with soil texture used in the study, we used the Gridded Soil Survey Geographic (gSSURGO) Database, which is an upgraded version based on the Soil Survey Geographic (SSURGO) Database. The database in gSSURGO includes 30 m resolution data that we used in our study (Soil Survey Staff, 2014). We described this in our manuscript in Lines 120-124.

In order to enhance soil moisture estimates in our future studies by integrating SMAP information, we added one sentence at the end of section 3.1 as:

"The daily soil temperature modeling could be further improved if high-resolution (e.g., 30 m and daily) satellite-based soil moisture/snow cover products become available, for example, products based on the SMAP or Sentinel satellites (Das et al., 2019)."

2. The second issue I see is with validation. Both the NRCS SCAN and Oklahoma MESONET sites don't report soil T at 10 cm (they are reporting at 4 and 8 cm for NRCS) and 4 cm for

MESONET. Please discuss how you use these data from different depths to train and validate a model that is at 10 cm?

**Responses**: When we downloaded soil temperature data from Oklahoma MESONET (http://www.mesonet.org/), we used the soil temperature data at 10 cm depth although OK-MESONET does include 5 cm, 10 cm, 25 cm, and 60 cm soil temperatures. Therefore, we directly used 10-cm soil temperature to train and test our models. Similarly, the data we selected from NRCS SCAN indicated they are 4-inch soil temperatures. We re-examined these descriptions and confirmed that they are 10 cm from NRCS SCAN (https://www.wcc.nrcs.usda.gov/scan/) for Texas.

Below are two screenshots from OK-MESONET and NRCS SCAN displaying soil temperature depths.

[Figure]

Here is the screenshot from the NRCS SCAN website

| Soil Moisture Percent -2in (pct) Mean of Hourly Values | Soil Moisture Percent -4in (pct) Mean of Hourly Values | Soil Moisture Percent -8in (pct) Mean of Hourly Values | Soil Moisture Percent -20in (pct) Mean of Hourly Values | Soil Moisture Percent -40in (pct) Mean of Hourly Values | Soil Temperature Observed -2in (degF) Mean of Hourly Values | Soil Temperature Observed -4in (degF) Mean of Hourly Values | Soil Temperature Observed -8in (degF) Mean of Hourly Values | Soil Temperature Observed -20in (degF) Mean of Hourly Values | Soil Temperature Observed -40in (degF) Mean of Hourly Values |
|---|---|---|---|---|---|---|---|---|---|
| | 49.7 | 19.7 | 17.8 | 16.4 | | 65 | 64 | 59 | 56 |
| | 47.2 | 25.3 | 17.9 | 16.4 | | 61 | 62 | 60 | 57 |
| | | 37.2 | 17.9 | 16.5 | | 59 | 59 | 59 | 57 |
| | | 35.7 | 17.8 | 16.5 | | 62 | 61 | 59 | 57 |
| | 50.4 | 34.4 | 17.9 | 16.5 | | 65 | 64 | 60 | 57 |
| | 46.0 | 33.1 | 17.9 | 16.6 | | 65 | 64 | 60 | 57 |
| | 42.4 | 31.8 | 17.9 | 16.6 | | 65 | 64 | 61 | 58 |

Specific comments:

1. Figs 5., 7, and 9: While RMSE is an important validation statistic, I would consider reporting other statistics such as BIAS and maybe the Nash-Sutcliffe Efficiency. RMSE really integrates both precision and accuracy while other statistics can help assess these independently.

**Response:** Thank you for your suggestions. We used the mean absolute error (MAE), which is the bias concept. Please see Line 202 in our original manuscript. Here we calculated Nash-Sutcliffe Efficiency (NSE) (see the Figure below) and found that we had similar or the same results as when we used RMSE.

[Figure]

**Figure R1.4**: Nash-Sutcliffe Efficiency (NSE) for the empirical model (EM02, **a**) and the improved model (iEM02, **b**). The colorbar defines values of NSE (-).

**--- The END of point-by-point response for referee #2**